# E-Cigarette-Only and Dual Use among Adolescents in Ireland: Emerging Behaviours with Different Risk Profiles

**DOI:** 10.3390/ijerph18010332

**Published:** 2021-01-05

**Authors:** Andrea K. Bowe, Frank Doyle, Debbi Stanistreet, Emer O’Connell, Michéal Durcan, Emmet Major, Diarmuid O’Donovan, Paul Kavanagh

**Affiliations:** 1Department of Public Health West, Health Service Executive, Merlin Park, Galway, Ireland; emeru.oconnell@hse.ie; 2Department of Health Psychology, Royal College of Surgeons in Ireland, Dublin 2, Ireland; FDoyle4@rcsi.ie; 3Department of Public Health and Epidemiology, Royal College of Surgeons in Ireland, Dublin 2, Ireland; Debbistanistreet@rcsi.ie (D.S.); paul.kavanagh@hse.ie (P.K.); 4Western Region Drug & Alcohol Task Force, Health Service Executive, Parkmore, Galway, Ireland; Micheal.Durcan2@hse.ie; 5Western Region Drug & Alcohol Task Force, Galway Roscommon Education Training Board, Parkmore Galway, Ireland; emmet.major@wrdatf.ie; 6School of Medicine, Dentistry and Biomedical Sciences, Centre for Public Health, Queens University, Belfast BT97BL, UK; D.ODonovan@qub.ac.uk; 7Institution: Health Intelligence, Strategic Planning and Transformation, Health Service Executive, Dublin 1, Ireland

**Keywords:** electronic nicotine delivery systems risk factors, tobacco products, adolescent, dual-use

## Abstract

E-cigarette-only use and dual-use are emerging behaviours among adolescent nicotine product users which have not yet been sufficiently explored. This study examines the prevalence of, and the factors associated with, nicotine product use in adolescence. The study is a cross-sectional analysis of the 2018 Planet Youth survey completed by 15–16 year olds in the West of Ireland in 2018. The outcome of interest was current nicotine product use, defined as use at least once in the past 30 days. A main effects multinomial logistic regression model was used to examine the association between potential risk and protective factors and nicotine product use. Among 4422 adolescents 22.1% were current nicotine product users, consisting of 5.1% e-cigarette only users, 7.7% conventional cigarette only users, and 9.3% dual-users. For risk factors, the odds of association were weaker for e-cigarette only use compared to conventional cigarette and dual use. Participating in team sport four times/week or more significantly reduced the odds of conventional cigarette and dual use but had no association with e-cigarette only use (Cig: adjusted odds ratio (AOR) 0.63, 95% confidence interval (CI) 0.44–0.90; Dual-use: AOR 0.63, 95% CI 0.43–0.93). Similarly, having higher value for conventional social norms reduced the odds of conventional cigarette and dual use but not e-cigarette only use. This is the first study to show, among a generalisable sample, that dual-use is the most prevalent behaviour among adolescent nicotine product users in Ireland. Risk factor profiles differ across categories of use and prevention initiatives must be cognisant of this.

## 1. Introduction

Tobacco use, nationally and globally, is a leading cause of preventable death and is one of the largest contributors to global disability adjusted life years [1,2]. It is estimated that 90% of adult smokers smoke their first cigarette before the age of 19 [3,4]. Initiation into tobacco use in adolescence risks early nicotine addiction, and an extended period of exposure to tobacco across the life course increases the risk of all smoking-related illness [5,6]. Over the past two decades, there has been significant progress made in developed countries in reducing youth tobacco use [7,8,9].

The e-cigarette was introduced to the European market in 2006 [7]. For established adult tobacco users with long standing nicotine addiction, the e-cigarette may be an effective harm reduction tool, a role which merits rigorous investigation for the benefits it could reap in this population [7,8,9]. However, this argument is not relevant to adolescent users, who receive all the pernicious properties of nicotine without the proposed benefits of harm reduction [3].

In the adolescent brain, nicotine exposure is associated with cognitive deficits, impairment in memory and in executive function [10]. Evidence that e-cigarettes lead to later tobacco use, a process known as the “gateway” effect, continues to mount [11]. Emerging literature suggests that the long term adverse effects include myocardial infarction [12], stroke [13], respiratory disease [14], and potential carcinogenic effects [15].

The prevalence of e-cigarette use among youth continues to grow, amid concerns that e-cigarette companies are employing the tactics traditionally used by tobacco companies to target youth users [16]. In 2019, the prevalence of current e-cigarette use, defined by use within the past 30 days, was 27.5% among high school students in the U.S. [17].

Measurement of e-cigarette use varies between countries making international comparisons challenging. In the United Kingdom for example, regular use, defined as using e-cigarettes at least weekly is more frequently measured. In 2019, regular use of e-cigarettes in the UK was reported by 1.6% of 11–18 year olds, an increase from 0.5% in 2015 [18].

In Ireland, the first nationally representative survey assessing e-cigarette use among 16–17 year olds in 2014 reported a prevalence of 3.2% for current use [19]. More recently, in 2018 among a cohort of 12–17 year olds, the prevalence of current use was 10% among boys and 7% among girls [20].

There are many limitations in epidemiological data on e-cigarette use. Standardising measurement for epidemiological surveys is challenging. E-cigarettes tend to be reusable devices, refilled with e-liquids which can vary in nicotine content. Unlike single-use cigarettes they cannot be easily counted, and frequency of use may vary from one puff at a time to continuous use throughout the day [21]. A further limitation is that often all e-cigarette users are categorised together. This category, however, consists of distinct user groups, those who use e-cigarettes only, and those who use both e-cigarettes and conventional cigarettes, referred to as dual-users.

Dual-use is an under-studied behaviour among adolescents. In an adult, dual-use may represent motivation to quit combustible cigarettes, however in adolescence, one would have to consider it unlikely to represent use as a smoking cessation aid [22,23]. E-cigarette use, in settings where conventional cigarettes are prohibited and would otherwise be smoke free, may increase overall nicotine exposure. The reasons underlying dual-use in adolescence have not yet been sufficiently explored.

Similarly, e-cigarette only use is another emerging behaviour requiring further investigation. Previous literature suggests dual-use tends to be strongly associated with factors considered to be traditional risk factors for conventional cigarette use. However e-cigarette only use tends to lack an association with many of these traditional risk factors [23,24,25,26]. A limitation in the majority of studies comparing risk factors across these groups is that “ever-use” of the product was used to define the user group, which makes it difficult to disentangle influences on current and persistent usage.

The aim of this study was therefore to describe the epidemiology of nicotine product use categorised according to current e-cigarette only, conventional cigarette only, and dual-use among 15–16 year olds in the west of Ireland and examine and compare the family, peer-group, and individual factors associated with these behaviours.

## 2. Materials and Methods

### 2.1. Study Design

This study is based on a cross-sectional analysis of data collected in October 2018 as part of the Planet Youth Pilot Programme in the West of Ireland [27].

### 2.2. Setting

The Planet Youth West 2018 Survey was undertaken in Galway, Mayo, and Roscommon, three counties in the West of Ireland. The population of this area as a whole was 453,400 in the most recent national census in 2016 [28].

All schools and Youthreach centres (Department of Education and Skills official programme for early school leavers) in these counties were invited to participate (*n* = 92).

### 2.3. Ethical Approval

Ethical approval was granted to the Planet Youth West study team to conduct these surveys by the Royal College of Physicians of Ireland. This study is based on analysis of the anonymised dataset.

### 2.4. Participants

The target population for the Planet Youth study were students in their fourth year of secondary school, aged 15–16 years old (*n* = 5729). Where school participation was agreed (*n* = 91), students and parents were given information about the survey and invited to opt out if they wished. Questionnaires were completed during school hours in October 2018. Data cleaning and processing was performed by the Icelandic Centre for Social Research and Analysis (ICSRA). As part of the data cleaning process forms which were insufficiently completed, referred to fictitious drugs or fictitious behaviours, or where the participant was aged ≥17 were eliminated (*n* = 316). There were a total of 4490 valid surveys from 15- and 16-year olds included in this study.

The West of Ireland has more people living in rural areas compared with the East of the country, however, comparing 2016 Census data for this geographic area with the Irish population as a whole suggests that they are similar across key sociodemographic variables, allowing a degree of confidence in generalising the results to the population as a whole [28].

### 2.5. Outcome

Current nicotine product use was defined by use of a nicotine product in the previous 30 days. Participants were asked “How many cigarettes, on average, have you smoked in the past 30 days?”. The seven response categories ranged from “none” to “more than 20 per day”. A response of none was categorised as “non-use” of that product and any other response was categorised as “current use of that product”. Participants were asked the same question for e-cigarettes. Participants were then categorised based on whether they were a current user or not of each product, into non-users, e-cigarette only users, conventional cigarette only users, and dual-users.

### 2.6. Predictors

The Icelandic Planet Youth prevention model is grounded in the classic sociological theory of social deviance [29]. From this theoretical perspective, adolescents are viewed as social products and not rational individual actors. Therefore, engagement in substance use is viewed as an attribute of the social environment the adolescent is exposed to, as opposed to an individual decision made by an adolescent [30]. As described by Kristjansson et al., the Planet Youth model, therefore, targets evidence-based domains within the adolescent social environment for intervention, namely family, the peer group, the school environment, and leisure time. The potential risk and protective factors included in this study were selected based on this theory and previous literature, and were categorised into the following groups:Sociodemographic factors: Gender, Parental Education;Family-related factors: Parental supervision, Parental smoking behaviour, Parental reaction to conventional cigarette use;Peer group-related factors: Friends smoking behaviours, Feeling it necessary to smoke to fit in;Community/Leisure time factors: Value for Conventional Social Norms, Team Sport Participation;Individual factors: Academic achievement, Self-rated Mental Health;

Parental supervision was measured based on the participant response to the following statements “My parents know who I am with in the evenings” and “My parents know where I am in the evenings” (two items, Cronbach’s alpha = 0.825). The four response categories ranged from “applies very poorly to me” to “applies very well to me” and were scored from 1 to 4 with higher scores indicating more parental supervision. This measure of parental supervision is consistent with previous published research using the Icelandic Planet Youth data [31].

Value for conventional social norms was measured based on response to eight items assessing commitment to social norms and attitude toward rules (eight items, Cronbach’s alpha 0.778). Examples include “You can break most rules if they don’t seem to apply,” “I follow whatever rules I want to follow”. The five response categories ranged from “strongly agree” to “strongly disagree” and were scored from 1 to 5 with higher scores indicating more value for conventional social norms. This measure of value for conventional social norms is derived from Dean 1961 [32] and has been used in previous published literature [33].

### 2.7. Statistical Analysis

Statistical analysis was performed using IBM SPSS Statistics 24. Descriptive analysis was performed using the cross-tabulations procedure and chi square statistic to detect significant differences in proportions. One-way ANOVA (analysis of variance) was used to compare means across groups.

A univariate analysis was performed for each variable. A main effects multinomial logistic regression including all variables was undertaken to examine the independent association between potential risk and protective factors and current nicotine product use. Listwise deletion was performed for the multinomial regression model and participants with missing data on any of the model variables were excluded from the model (*n* = 340). Results are presented as adjusted odds ratios (AOR), 95% confidence intervals and *p*-values. A *p*-value of less than 0.05 was deemed to be significant for all statistical tests.

## 3. Results

### 3.1. Description of Sample

There were 4490 15–16 year olds eligible for inclusion in this study. Figure 1 describes the participants included at each stage of the study.

### 3.2. Characteristics of Study Population

The characteristics of the study population are described in Table 1. Overall, 50.8% were female and 52% reported higher level maternal education. Among all participants 26.7% reported they had a parent who smoked and 70.8% reported they had at least a few friends who smoked.

The mean score for value for conventional social norms differed significantly between the groups and was significantly higher for e-cigarette only users. Participation in team sport ≥4 times/week was reported by 26.9% of e-cigarette only users, 24.8% of non-users, 19.4% of conventional cigarette users, and 17% of dual-users.

### 3.3. Prevalence of Nicotine Product Use

The prevalence and characteristics of nicotine product use are described in Table 2. Overall 22.1% were current nicotine product users. This consisted of 5.1% e-cigarette only use, 7.7% conventional cigarette only use, and 9.3% dual-use.

### 3.4. Results of Multinomial Logistic Regression

#### 3.4.1. Sociodemographic Factors

The results of the multinomial logistic regression model are shown in Table 3. Compared with females, males were twice as likely to be e-cigarette only or dual-users. There was no significant association between gender and conventional cigarette use.

On univariate analysis, primary level maternal education was significantly associated with increased odds of all nicotine product use, however on multivariate analysis significance was retained only for e-cigarette only use.

#### 3.4.2. Family-Related Factors

Having a parent who smokedincreased the odds of all nicotine product use. Compared to those who felt their parents would be strongly against their use of conventional cigarettes, for those who thought their parents less strongly against it, the odds of conventional cigarette use were 3.5 times higher and of dual-use 5 times higher. There was no significant association with e-cigarette only use.

A factor with a potentially protective association with nicotine product use was parental supervision, which had a significant inverse association with all behaviours. For every 1 standard deviation increase in the parental supervision score, the odds of e-cigarette only use were 30% lower, and of conventional cigarette and dual-use 40% lower.

#### 3.4.3. Peer-Related Factors

There was a strong and statistically significant association between peer-related factors and all nicotine product use. For those who felt it was necessary to smoke to fit in the odds of nicotine product use were higher across all categories. For those who reported almost all of their friends smoked the odds of e-cigarette only, conventional cigarette only and dual-use were 5, 40.5, and 31 times higher, respectively.

#### 3.4.4. Community/Leisure Time Related Factors

Increased value for conventional social norms was significantly associated with reduced odds of conventional cigarette and dual-use, but not e-cigarette only use. A 1 standard deviation increase in the value for conventional social norms corresponded to a 20% and 30% reduction in the odds of conventional cigarette and dual-use.

Similarly, participating in team sports ≥4 times/week was significantly associated with reduced odds of conventional cigarette and dual-use but not e-cigarette use. Compared to those who almost never participate, those involved in team sports ≥4 times/week were 40% less likely to be a conventional or dual-user, but no less likely to be an e-cigarette only user

#### 3.4.5. Individual Factors

Compared to those who reported above average academic performance, those who reported average or below had increased odds of nicotine product use. On univariable analysis poor self-reported mental health was associated with increased odds of conventional cigarette and dual-use but significance was not retained in the multivariable model.

## 4. Discussion

### 4.1. Epidemiology of Nicotine Product Use

This is the first study to show, in a generalisable sample covering the West of Ireland, that dual-use is the most prevalent behaviour among adolescent nicotine product users in Ireland. This study also shows that the prevalence of e-cigarette use among 15–16 year olds has increased 4.5 fold since 2014 [19].

Direct comparison with international data on adolescent e-cigarette use is challenging due to the use of differing survey methodologies, differing definitions of use, and differing age groups recruited to surveys. Therefore, one must be cautious in making international comparisons. The prevalence of e-cigarette use in this study is lower than the 28% reported among 14–18 year olds in the US [17], but higher than the 6% prevalence reported among a younger cohort of 11–15 year olds in England [34].

Regarding ever use of e-cigarettes, 40.1% of adolescents in this study reported ever use of e-cigarettes. This is higher than the prevalence of ever use of 34% reported among 17–18 year olds in the Growing up in Ireland study in 2016, and higher than the 25% reported by the NHS for 11–15 year olds in England [34,35]. It is comparable with the prevalence of ever use among high school students, typically age 14–18, in the USA which was 46.9% in 2019 [17].

Dual-use was the most prevalent behaviour among nicotine product users in this study. This is similar to the U.S where 10.8% of high school students report current use of two or more tobacco products but higher than other international comparisons using the same categorisation, such as a Korean study of 13–18 year olds where dual-use was reported by 4.9% [17,26]. Dual-use of e-cigarettes and conventional cigarettes is not routinely measured in many surveys.

Dual-use is a concerning behaviour among adolescents and one that is understudied. If e-cigarettes are being used in settings where smoking is prohibited, this may increase cumulative nicotine exposure and the associated adverse effects. It is possible that dual-use may represent a transition to tobacco product use, a process known as the gateway effect, and longitudinal evidence for this effect is mounting [36,37].

### 4.2. Factors Associated with Nicotine Product Use

The strength of association between “traditional” risk and protective factors tended to be weakest for e-cigarette only users and stronger for conventional cigarette and dual-users, a finding consistent with previous literature in this area [23,24,25,26]. These findings add weight to concerns that e-cigarettes may be recruiting lower risk adolescents who would not otherwise engage in conventional smoking.

A notable finding in this study was the importance of the perceived parental reaction to the specific nicotine product. Almost all participants (94%) perceived their parents to be strongly against conventional cigarettes. Adolescents who believed their parents were strongly against conventional cigarettes had reduced odds of conventional cigarette use but not e-cigarette use.

The public health message around e-cigarettes has been mixed—Public Health England promote a message that they are 95% safer than conventional cigarettes [8]. This message is applicable to adult smokers with an established nicotine addiction, using them as a harm reduction tool. For most adolescents, e-cigarettes are not being used as an adjunct to quit a sustained smoking habit. Therefore, adolescents are being exposed to the harmful effects of e-cigarettes without the potential benefits of harm reduction. The public health message must be tailored accordingly. An unambiguous public health message on the harms of e-cigarette use among adolescents is required.

In this study, there was a statistically significant inverse association between value for conventional social norms and conventional cigarette and dual-use. Those with higher value for conventional social norms were less likely to be conventional cigarette or dual-users but were not less likely to be e-cigarette only users. The lack of association with e-cigarette use is concerning and supports the hypothesis that e-cigarette use is viewed as a more socially acceptable behaviour, one that, unlike conventional smoking, society does not attach the same negative connotations to.

A finding which further supports this concern is in relation to team sport participation. This study found that participation in team sport was significantly associated with reduced odds of conventional cigarette and dual-use. No such association was found for e-cigarette use. This finding is in keeping with previous literature. A study of more than 15,000 U.S. adolescents in 2016 found adolescents who participated in three or more competitive sports were 40% less likely to be a conventional smoker and 50% less likely to be a dual-user, however, no association was found between competitive sport and e-cigarette use [38].

The protective influence of sport participation on conventional smoking may be due to the widely accepted negative health impacts of conventional smoking among a population who place high value on health; or may be due to conventional smoking being a non-acceptable behaviour among athletes. E-cigarettes, marketed as a healthier alternative, may appeal to those engaged in healthy behaviours such as sport. These findings again support the hypothesis that e-cigarettes are targeting a lower risk group who would not otherwise engage in smoking.

### 4.3. Strengths and Limitations

The strengths of this study include its large sample size, high response rate, and generalisability. The study is the first Irish study to compare and contrast potential risk and protective factors across e-cigarette, conventional cigarette, and dual-users. Unlike other international studies which have used ‘ever use’ as an outcome measure, this study examined factors associated with ‘current use’ which is a more informative measure in guiding policy.

There are however limitations to this study, perhaps the most significant being that it is cross sectional. Therefore, it is not possible to establish a temporal relationship between exposure and outcome and not possible to infer causation. The factors are referred to in the study as potential risk and protective factors. Sociodemographic information was not available for those who did not respond to the parent study and they may systematically differ from those who did. Previous studies have found nonresponders are often more likely to be those who would report risk behaviours which can therefore lead to underestimation of risk factor prevalence.

The surveys were completed during school hours and on school premises, which may introduce bias in the reporting of behaviours, particularly nicotine product use. They were also self-reported and are therefore subject to recall bias. A school identifier was not included in the dataset in order to ensure anonymity meaning that clustering of behaviours within schools could not be measured in this study.

Due to collinearity maternal education was chosen as the single marker of socioeconomic status. However, using a single marker of socioeconomic status is not without limitation and residual confounding due to socioeconomic status may exist in the model.

The Planet Youth questionnaire did not decipher between nicotine containing e-cigarette use and non-nicotine containing e-cigarette use. Previous literature in this area suggests that adolescents are often unaware of whether the e-liquid/e-cigarette they use contains nicotine [39]. Further questions to distinguish between these products could be considered for future surveys.

## 5. Conclusions

In conclusion, in a generalisable sample of adolescents in the West of Ireland, 9.3% of adolescents reported dual-use of e-cigarettes and conventional cigarettes in the past 30 days. This was the most prevalent behaviour among nicotine product users. Current e-cigarette use was reported by 14.4% of adolescents and this represents a 4.5-fold increase since 2014 when the prevalence among a similar age-cohort was 3.2%. Prior to 2014, e-cigarette use had not been measured among Irish adolescents. Over a 4-year period, Ireland has seen a rapid increase in e-cigarette use among youth, and is not alone in this observation. In the United States where e-cigarette use is systematically monitored on an annual basis, current e-cigarette use among high school students increased 9-fold between 2011 and 2015 [17].

This study found that the risk factor profiles of e-cigarette only users differ from that of conventional cigarette and dual-users.

Findings of this study and others support the hypothesis that e-cigarettes, marketed as a healthier and more socially acceptable alternative, may appeal to a population otherwise engaged in positive lifestyle behaviours who would not use conventional cigarettes. The use of flavourings and advertising targeting youth is being employed in a manner almost identical to that seen at the beginning of the tobacco epidemic. The public health community, policy makers, and the public must act now. Individual and community prevention efforts must acknowledge the unique risk factor profiles of adolescent nicotine product users. Legislative changes are urgently required in Ireland to regulate advertising, particularly in the online environment, and to restrict flavourings which undeniably target youth users.

## Figures and Tables

**Figure 1 ijerph-18-00332-f001:**
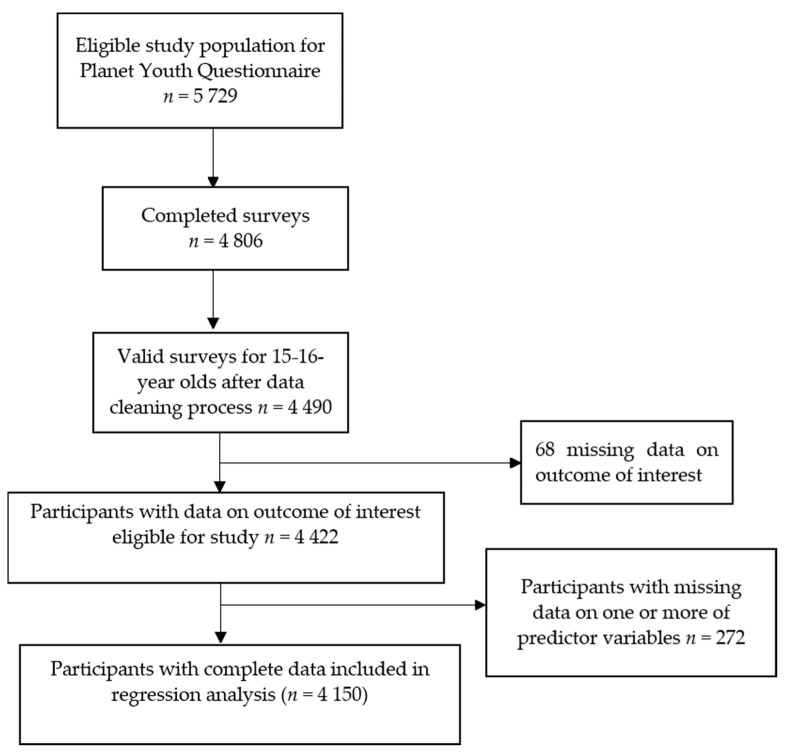
Flowchart of study population showing number of participants included in descriptive epidemiology (*n* = 4422) and number included in regression model (*n* = 4150).

**Table 1 ijerph-18-00332-t001:** Description of study population (*n* = 4422).

	Valid	Total	Non-User	E-Cig Only	Cig Only	Dual-User	*p*
		4422	3447	225	339	411	
**Sociodemographic**	***n***	***n* (%)**					
Gender	4407						
Male		2170 (49.2%)	1596 (46.5%)	155 (68.9%)	164 (48.5%)	255 (62.3%)	
Female		2237 (50.8%)	1839 (53.5%)	70 (31.1%)	174 (51.5%)	154 (37.7%)	<0.001
Maternal education	4409						
Primary		308 (7%)	199 (5.8%)	23 (10.3%)	39 (11.6%)	47 (11.5%)	
Secondary		1121 (25.4%)	858 (24.9%)	57 (25.4%)	84 (25%)	122 (29.8%)	
Tertiary		2291 (52.0%)	1874 (54.5%)	89 (39.7%)	159 (47.3%)	169 (41.3%)	
Did not know		689 (15.6%)	509 (14.8%)	55 (24.6%)	54 (16.1%)	71 (17.4%)	<0.001
**Family-related factors**							
Parent smokes	4397						
Yes		1172 (26.7%)	766 (22.3%)	82 (36.8%)	127 (37.7%)	197 (48.9%)	
No		3225 (73.3%)	2668 (77.7%)	141 (63.2%)	210 (62.3%)	206 (51.1%)	<0.001
Parent reaction to cig use	4374						
Totally/very against		4113 (94%)	3315 (97.2%)	212 (94.6%)	284 (84.3%)	302 (75.3%)	<0.001
Rather/would not care		261 (6.0%)	97 (2.8%)	12 (5.4%)	53 (15.7%)	99 (24.7%)	
Parental supervision	4422						
Mean (SD)		6.92 (1.39)	7.17 (1.19)	6.42(1.52)	6.14(1.58)	5.74(1.78)	<0.001
**Peer-related factors**							
Smoke to fit in	4414						
Yes		610 (13.8%)	261 (7.6%)	43 (19.2%)	125 (36.9%)	181 (44.3%)	
No		3804 (86.2%)	3181 (92.4%)	181 (80.8%)	214 (63.1%)	228 (55.7%)	<0.001
Friends who smoke	4354						
None		1269 (29.1%)	1213 (35.7%)	33 (14.9%)	9 (2.7%)	14 (3.5%)	
A few/some		2487 (57.1%)	1952 (57.5%)	141 (63.5%)	218 (64.7%)	176 (44.2%)	
Most/almost all		598 (13.7%)	232 (6.8%)	48 (21.6%)	110 (32.6%)	208 (52.3%)	<0.001
**Community/leisure**							
Value for social norms	4398						
Mean (SD)		22.20 (5.79)	22.94 (5.66)	21.47(5.48)	19.70(5.31)	18.40(5.38)	<0.001
Team sport participation	4351						
Almost never		1599 (36.8%)	1194 (35.1%)	64 (29.2%)	163 (48.7%)	178 (44.6%)	
1–3 times/week		1716 (39.4%)	1360 (40.0%)	96 (43.8%)	107 (31.9%)	153 (38.3%)	
≥4 times/week		1036 (23.8%)	844 (24.8%)	59 (26.9%)	65 (19.4%)	68 (17.0%)	<0.001
**Individual**							
Academic achievement	4414						
Below average		439 (9.9%)	248 (7.2%)	42 (18.7%)	51 (15.0%)	98 (23.8%)	
Average		1730 (39.2%)	1275 (37.1%)	96 (42.7%)	165 (48.7%)	194 (47.2%)	
Above average		2245 (50.9%)	1916 (55.7%)	87 (38.6%)	123 (36.3%)	119 (29.0%)	<0.001
Mental health	4408						
Good or very good		2626 (59.6%)	2165 (62.9%)	131 (58.2%)	152 (45.1%)	178 (44.0%)	
Moderate		1058 (24.0%)	796 (23.1%)	57 (25.3%)	98 (29.1%)	107 (26.4%)	
Bad or very bad		724 (16.4%)	480 (13.9%)	37 (16.4%)	87 (25.8%)	120 (29.6%)	<0.001

Maternal Education: Tertiary = graduated from a university/technical college/apprenticeship, Secondary = graduated from secondary school or started but did not finish university/technical college/apprenticeship, Primary = primary school or less completed, started secondary but did not finish. *p* = *p*-value.

**Table 2 ijerph-18-00332-t002:** Smoking behaviours of 15–16 year olds in the West of Ireland (*n* = 4412).

	Valid	Total	Male	Female	*p* Value
	*n* (% of Total)		*n* (% of Males)	*n* (% of Females)	
Current nicotine product user	4412	977 (22.1%)	577 (26.6%)	400 (17.9%)	<0.001
Ever nicotine product user	4419	2160 (48.9%)	1173 (53.9%)	987 (44.0%)	<0.001
Conventional Cigarettes					
Ever use	4410	1652 (37.5%)	870 (40.0%)	782 (35.0%)	
Current use	4412	749 (17.0%)	420 (19.3%)	329 (14.7%)	<0.001
Electronic Cigarettes					
Ever use	4424	1776 (40.1%)	1010 (46.4%)	766 (34.1%)	
Current use	4412	637 (14.4%)	412 (18.9%)	225 (10.0%)	<0.001
Current Nicotine Product Use	4407				
None		3435 (77.9%)	1596 (73.5%)	1839 (82.2%)	
E-Cigarette only		225 (5.1%)	155 (7.1%)	70 (3.1%)	
Conventional cigarette only		338 (7.7%)	164 (7.6%)	174 (7.8%)	
Dual-use		409 (9.3%)	255 (11.8%)	154 (6.9%)	<0.001
Ever Nicotine Product Use	4407				
None		2259 (51.3%)	1005 (46.3%)	1254 (56.1%)	
E-cigarette only		498 (11.3%)	298 (13.7%)	200 (8.9%)	
Conventional cigarette only		382 (8.7%)	161 (7.4%)	221 (9.9%)	
Dual-use		1268 (28.8%)	707 (32.6%)	561 (25.1%)	<0.001

**Table 3 ijerph-18-00332-t003:** Multinomial regression model examining the association between potential risk and protective factors and nicotine product use.

	E-Cig vs. Non				Cig vs. Non				Dual Use vs. Non			
	*n* (%)	AOR	95% CI	*p*	*n* (%)	AOR	95% CI	*p*	*n* (%)	AOR	95% CI	*p*
**Sociodemographic**												
Gender												
Female	66 (31.7)	Ref			165 (51.6)				143 (39.0)			
Male	142 (68.3)	2.24	1.61–3.10	<0.001	155 (48.4)	1.28	0.97–1.68	0.081	224 (61.0)	2.36	1.76–3.16	<0.001
Maternal education												
Tertiary	82 (39.4)	Ref			154 (48.1)				151 (41.1)			
Secondary	55 (26.4)	1.18	0.82–1.71	0.370	79 (24.7)	0.82	0.60–1.12	0.209	109 (29.7)	1.08	0.78–1.49	0.635
Primary	22 (10.6)	1.89	1.12–3.19	0.016	37 (11.6)	1.42	0.90–2.33	0.133	44 (12.0)	1.45	0.91–2.31	0.122
Did not know	49 (23.6)	1.40	0.94–2.07	0.100	50 (15.6)	0.76	0.52–1.11	0.149	63 (17.2)	0.79	0.54–1.15	0.218
**Family-related factors**												
Parent smokes												
No	131 (63.0)	Ref			201 (62.8)				184 (50.1)			
Yes	77 (37.0)	1.71	1.25–2.34	<0.001	119 (37.2)	1.58	1.20–2.09	0.001	183 (49.9)	2.44	1.85–3.22	<0.001
Parental reaction to cigarette use												
Totally/very against	196 (94.2)	Ref			272 (85.0)				278 (75.7)			
Rather against/would not care	12 (5.8)	1.31	0.69–2.51	0.409	48 (15.0)	3.49	2.27–5.37	<0.001	89 (24.3)	4.65	3.09–7.01	<0.001
Parental supervision: mean (SD)												
1 SD increase corresponds to	7 (2)	0.71	0.62–0.82	<0.001	6 (3)	0.63	0.56–0.71	<0.001	6 (2)	0.60	0.53–0.67	<0.001
**Peer-related factors**												
Feel necessary to smoke to fit in												
No	168 (80.8)	Ref			203 (63.4)				213 (58.0)			
Yes	40 (19.2)	2.13	1.45–3.13	<0.001	117 (36.6)	4.36	3.25–5.83	<0.001	154 (42.0)	5.13	3.81–6.91	<0.001
Friends who smoke												
None	30 (14.4)	Ref			7 (2.2)				13 (3.5)			
A few/some	131 (63)	2.15	1.45–3.13	<0.001	206 (64.4)	14.19	6.60–30.53	<0.001	162 (44.1)	5.41	3.0–9.76	<0.001
Most/almost all	47 (22.6)	5.19	3.10–8.44	<0.001	107 (33.4)	40.52	18.31–89.68	<0.001	192 (52.3)	31.44	17.06–57.92	<0.001
**Community/leisure**												
Value for conventional social norms: mean (SD)												
1 SD increase corresponds to	21 (7)	0.95	0.81–1.12	0.535	19 (7)	0.78	0.66–0.91	0.001	18 (7)	0.68	0.57–0.79	<0.001
Team sport involvement												
Almost never	59 (28.4)				156 (48.8)				162 (44.1)			
1–3 times/week	93 (44.7)	1.34	0.94–1.92	0.105	102 (31.9)	0.57	0.43–0.77	<0.001	142 (38.7)	0.77	0.57–1.05	0.094
≥4 times/week	56 (26.9)	1.31	0.87–1.97	0.199	62 (19.4)	0.63	0.44–0.90	0.011	63 (17.2)	0.63	0.43–0.93	0.019
**Individual**												
Self-reported academic achievement												
Above average	80 (38.5)				116 (36.3)				106 (28.9)			
Average	89 (42.8)	1.43	1.03–1.98	0.03	158 (49.4)	1.58	1.20–2.09	0.001	175 (47.7)	1.76	1.30–2.37	<0.001
Below average	39 (18.8)	2.53	1.62–3.93	<0.001	46 (14.4)	1.47	0.96–2.26	0.076	86 (23.4)	2.43	1.62–3.63	<0.001
Self-reported mental health												
Good or very good	120 (57.7)				143 (44.7)				158 (43.1)			
Moderate	51(24.5)	1.04	0.72–1.48	0.853	93 (29.1)	1.13	0.83–1.54	0.44	99 (27.0)	1.06	0.76–1.47	0.723
Bad or very bad	37 (17.8)	1.16	0.75–1.78	0.506	84 (26.3)	1.22	0.85–1.74	0.275	110 (30.0)	1.32	0.92–1.89	0.130

AOR: adjusted odds ratio. SD: standard deviation. 95% CI = 95% confidence interval. *p* = *p* value. E-cig = e-cigarette. Cig = conventional cigarette. Non = not a current user of nicotine products. Valid 4150 Nagelkerke r^2^ = 0.400.

## Data Availability

Not applicable.

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
