# Peer review of "E-Cigarette-Only and Dual Use among Adolescents in Ireland: Emerging Behaviours with Different Risk Profiles"

_ijerph, 2021, doi:10.3390/ijerph18010332_

Round 1
Reviewer 1 Report
It is very interesting research, just two comments for discussion:
1. Include questions about heated tobacco products and flavored capsule cigars for future research. These products are very attractive to teenagers.
2. It is proposed to carry out an analysis where the results of previous studies on the protective and risk factors only for the use of tobacco are compared with those obtained in the study that is presented, to know if they have changed over time.
Author Response
Dear Reviewer 1,
Thank you for reviewing this paper and thank you for the points you have raised. Please find response to each point below.
1.Include questions about heated tobacco products and flavored capsule cigars for future research. These products are very attractive to teenagers.
Response: Agreed. At the time this survey was carried out in Ireland in 2018, electronic cigarettes appeared to be the most prevalent nicotine product (other than conventional cigarettes) being used. The Planet Youth 2018 Questionnaire asked specifically about electronic cigarette use.
However since then other products, as you have pointed out, have continued to emerge such as heated tobacco products. The Planet Youth Questionnaire for this year has been agreed but additional questions in response to other emerging nicotine products could be considered for future surveys.
2. It is proposed to carry out an analysis where the results of previous studies on the protective and risk factors only for the use of tobacco are compared with those obtained in the study that is presented, to know if they have changed over time.
Thank you for this suggestion for further research. We agree that this would be an interesting comparison and will consider this for future research. The Planet Youth study will be conducted again in the West of Ireland in 2020 and in 2022 and this will also allow for analysis of trends in use and for comparison of changes in risk and protective factors.
Reviewer 2 Report
This is a welcome paper on the phenomenon of dual-use in adolescents. A few comments re. this overall sound and timely piece
a) Page 2/ lines 99-100: May need to define West of Ireland area (counties, population) for the audience.
b) Page 2/ Lines 109-110: Why is the survey administered to this narrow age group of fourth-year students? Many adolescent tobacco use surveys go broader (e.g. 15-18 years, middle schoolers, grades 10/11/12). The authors might wish to explain why the exact age 15-16 is of interest rather than a wider definition of teens/adolescents/children. Additionally, what is the total population of 15-16 year olds in the West of Ireland?
c) Tables 1 and 2 : with any cross-sectional survey, it is useful for readers to be able to compare with a recent/ representative survey, e.g a national telephone or school-based survey, even if all questions and cross-tabs or years do not line up. Could the authors try to incorporate and compare to the statistics from a reference survey (in addition to the discussion they have in section 4.1) ?
d) (Throughout)- e-cigarettes are taken to be synonymous with nicotine-containing e-cigarettes. Given the proliferation of types of products (refillable, disposable, nicotine and non-nicotine containing) is there an archetypal e-cigarette students were queried about? Is there any way to confirm that e-cigarettes smoked are always nicotine containing (e.g. third party market study for Ireland or such data?)
e) Section 4.2, lines 51-53: The observation about mixed messaging is well-taken. The authors might wish to add that to their conclusion - a 22 % youth prevalence is high and for public health advocacy, the finding that most use is e-cigarette use is quite stark.
f) Additional commentary (e.g in the conclusion ) on how rapidly this prevalence increase has happened might be salutory here - for example through a comparison to any pre-2015 surveys. The proliferation of youth ecigarette use in the US and other places has happened just when the public health community was celebrating long term declines in youth prevalence, and this paper on Ireland only highlights how stark the sudden normalization of e-cigarette use has been.
Author Response
Dear Reviewer 2,
Thank you for reviewing this research and thank you for your comments and suggestions below. Please find detailed response to each point below.
a) Page 2/ lines 99-100: May need to define West of Ireland area (counties, population) for the audience.
Response: We agree for an international audience a definition of the geographic area is helpful. We have not added the lines "The Planet Youth West 2018 Survey was undertaken in Galway, Mayo and Roscommon, three counties in the West of Ireland. The population of this area as a whole was 453,400 in the most recent national census in 2016".
b) Page 2/ Lines 109-110: Why is the survey administered to this narrow age group of fourth-year students? Many adolescent tobacco use surveys go broader (e.g. 15-18 years, middle schoolers, grades 10/11/12). The authors might wish to explain why the exact age 15-16 is of interest rather than a wider definition of teens/adolescents/children. Additionally, what is the total population of 15-16 year olds in the West of Ireland?
Response: he total population of 15-16 year olds post Junior Certificate in the West of Ireland was 5,729 (Junior certificate is the state exam which all students sit). The legal age for leaving school in Ireland is 16 so the vast majority of all students will sit this exam.
The Planet Youth questionnaire examines many aspects of adolescent life, of which tobacco/e-cigarette use is one. It also examines mental health, family life, drug use, sport and activity. The Planet Youth programme was originally developed in Iceland and was initiated among 15-16 year olds. This is the reason this group was chosen for the Planet Youth Pilot programme in the West.
c) Tables 1 and 2 : with any cross-sectional survey, it is useful for readers to be able to compare with a recent/ representative survey, e.g a national telephone or school-based survey, even if all questions and cross-tabs or years do not line up. Could the authors try to incorporate and compare to the statistics from a reference survey (in addition to the discussion they have in section 4.1) ?
Response: Agree that this comparison can provide useful information for readers. However, to date the literature on e-cigarette use in Ireland has been collected among different age groups, using different methods of ascertainment and there is no comparable national survey which has measured e-cigarette only and dual-use in the way measured in this study. However the Planet Youth West 2020 survey is currently underway (and will be repeated again in 2022) and this will allow us to perform a direct comparison using the same age group and the same ascertainment in future research.
d) (Throughout)- e-cigarettes are taken to be synonymous with nicotine-containing e-cigarettes. Given the proliferation of types of products (refillable, disposable, nicotine and non-nicotine containing) is there an archetypal e-cigarette students were queried about? Is there any way to confirm that e-cigarettes smoked are always nicotine containing (e.g. third party market study for Ireland or such data?)
Response: This is an important point raised. The questionnaire asked about "e-cigarette use" but did not further define or specify whether these e-cigarettes contained nicotine or not. There is limited data on e-cigarette use in Ireland and no published results examining this particular element. We acknowledge that this is a limitation in the questioning on e-cigarette use and that future surveys may benefit from including further questions on nicotine content. We have added the following lines to the discussion of limitations of the study
"The Planet Youth questionnaire did not decipher between nicotine containing e-cigarette use and non-nicotine containing e-cigarette use. Previous literature in this area suggests that adolescents are often unaware of whether the e-liquid/e-cigarette they use contains nicotine [39]. Further questions to distinguish between these products could be considered for future surveys".
e) Section 4.2, lines 51-53: The observation about mixed messaging is well-taken. The authors might wish to add that to their conclusion - a 22 % youth prevalence is high and for public health advocacy, the finding that most use is e-cigarette use is quite stark.
f) Additional commentary (e.g in the conclusion ) on how rapidly this prevalence increase has happened might be salutory here - for example through a comparison to any pre-2015 surveys. The proliferation of youth ecigarette use in the US and other places has happened just when the public health community was celebrating long term declines in youth prevalence, and this paper on Ireland only highlights how stark the sudden normalization of e-cigarette use has been.
Response to (e) and (f) above. We agree with these suggestions and have added the following paragraph to the conclusions
"Prior to 2014, e-cigarette use had not been measured among Irish adolescents. Over a four-year period, Ireland has seen a rapid increase in e-cigarette use among youth, and is not alone in this observation. In the United States where e-cigarette use is systematically monitored on an annual basis, current e-cigarette use among high school students increased by 900% between 2011 and 2015 [17]."
Reviewer 3 Report
The followings are suggestions:
Line 50: ... increasing the risk of all smoking-related illness [5,6]. Over the past two decades, there has been
Suggestion: ... increase the risk of all smoking-related illness [5,6]. Over the past two decades, there has been
Line 64-65: ... [16]. In the U.S. there was a nine-fold increase in past 30 day e-cigarette use among high school 64 students between 2011 and 2015.
Suggestion: (not sure how the red phrase fit in there. Past 30 days from when? Please read and adjust. The whole sentence did not flow please review.)
Line 102: All schools and Youthreach centres [Department of Education and Skills official programme
Suggestion: Not sure if this is a new vocabulary or name of a program/center. please compare "youthreach" versus "youth outreach" and decide the best to use.
Line 109: The target population for the Planet Youth study was students in their fourth year of secondary
Suggestion: check grammar "was" or "were" which is most appropriate?
Line 117: The West of Ireland has more people living rural areas compared with the East of the country,
Suggestion: The West of Ireland has more people living in rural areas compared with the East of the country,
Line 175-176: Figure 1. Flowchart of Study Population Showing Number of Participants Included in Descriptive 175 Epidemiology (n=4,422) and Number Included in Regression Model (n=4,150).
suggestion : The caption should be under the figure and instead a title is put on top of the figure
Author Response
Dear Reviewer 3,
Thank you for reviewing this research and for your corrections and suggestions below. Please find detailed responses to each point below.
Line 50: ... increasing the risk of all smoking-related illness [5,6]. Over the past two decades, there has been
Suggestion: ... increase the risk of all smoking-related illness [5,6]. Over the past two decades, there has been
Response: This has been changed accordingly.
Line 64-65: ... [16]. In the U.S. there was a nine-fold increase in past 30 day e-cigarette use among high school 64 students between 2011 and 2015.
Suggestion: (not sure how the red phrase fit in there. Past 30 days from when? Please read and adjust. The whole sentence did not flow please review.)
Response: This sentence has now been adjusted to
"In 2019, the prevalence of current e-cigarette use, defined by use within the past 30 days, was 27.5% among high school students in the U.S. [17]".
The reference to the rapid nine-fold increase is now included in the conclusions section.
Line 102: All schools and Youthreach centres [Department of Education and Skills official programme
Suggestion: Not sure if this is a new vocabulary or name of a program/center. please compare "youthreach" versus "youth outreach" and decide the best to use.
Response: This term Youthreach refers to the name of a program in Ireland. We believe this does not require quotation marks but requires capitalisation.
Line 109: The target population for the Planet Youth study was students in their fourth year of secondary
Suggestion: check grammar "was" or "were" which is most appropriate?
Response: This has been changed to were.
Line 117: The West of Ireland has more people living rural areas compared with the East of the country,
Suggestion: The West of Ireland has more people living in rural areas compared with the East of the country,
Response. This has been changed accordingly.
Line 175-176: Figure 1. Flowchart of Study Population Showing Number of Participants Included in Descriptive 175 Epidemiology (n=4,422) and Number Included in Regression Model (n=4,150).
suggestion : The caption should be under the figure and instead a title is put on top of the figure
Response: This has been changed accordingly.